# Influence of ZnO Nanoparticles on the Properties of Ibuprofen-Loaded Alginate-Based Biocomposite Hydrogels with Potential Antimicrobial and Anti-Inflammatory Effects

**DOI:** 10.3390/pharmaceutics15092240

**Published:** 2023-08-30

**Authors:** Delia Mihaela Rata, Anca Niculina Cadinoiu, Oana Maria Daraba, Luiza Madalina Gradinaru, Leonard Ionut Atanase, Daniela Luminita Ichim

**Affiliations:** 1Faculty of Medical Dentistry, “Apollonia” University of Iasi, Pacurari Street, No. 11, 700511 Iasi, Romania; iureadeliamihaela@yahoo.com (D.M.R.); leonard.atanase@yahoo.com (L.I.A.); danielaluminitaichim@yahoo.com (D.L.I.); 2“Petru Poni” Institute of Macromolecular Chemistry, 41A Grigore Ghica Voda Alley, 700487 Iasi, Romania; gradinaru.luiza@icmpp.ro; 3Academy of Romanian Scientists, 050045 Bucharest, Romania

**Keywords:** hyaluronic acid, alginate, hydrogels, ZnO nanoparticles, Ibuprofen, controlled drug release, in vitro tests

## Abstract

Hydrogels are a favorable alternative to accelerate the burn wound healing process and skin regeneration owing to their capability of absorbing contaminated exudates. The bacterial infections that occur in burn wounds might be treated using different topically applied materials, but bacterial resistance to antibiotics has become a major problem worldwide. Therefore, the use of non-antibiotic treatments represents a major interest in current research. In this study, new antibiocomposite hydrogels with anti-inflammatory and antimicrobial properties based on hyaluronic acid (HA) and sodium alginate (AG) were obtained using 4-(4,6-dimethoxy-1,3,5-triazinyl-2)-4-methylmorpholinium chloride as an activator. The combination of Ibuprofen, a non-steroidal anti-inflammatory drug commonly used to reduce inflammation, fever and pain in the body, with zinc oxide nanoparticles (ZnO NPs) was used in this study aimed at creating a complex hydrogel with anti-inflammatory and antimicrobial action and capable of improving the healing process of wounds caused by burns. FTIR spectra confirmed the cross-linking of AG with HA as well as the successful incorporation of ZnO NPs. Using electronic microscopy, it was noticed that the morphology of hydrogels is influenced by the incorporation of ZnO nanoparticles. Moreover, the incorporation of ZnO nanoparticles into hydrogels also has an influence on the swelling behavior at both pH 7.4 and 5.4. In fact, the swelling rate is lower when the amounts of the activator, HA and ZnO NPs are high. A drug release rate of almost 100% was observed for hydrogels without ZnO NPs, whereas the addition of nanoparticles to hydrogels led to a decrease in the release rate to 68% during 24 h. Cellular viability tests demonstrated the non-cytotoxic behavior of the hydrogels without the ZnO NPs, whereas a weak to moderate cytotoxic effect was noticed for hydrogels with ZnO NPs. The hydrogels containing 4% and 5% ZnO NPs, respectively, showed good antimicrobial activity against the *S. aureus* strain. These preliminary data prove that these types of hydrogels can be of interest as biomaterials for the treatment of burn wounds.

## 1. Introduction

The skin is considered a large multifunctional organ that has several important roles, for example, it protects the internal organs from the environment and external factors, maintains homeostasis and regulates temperature [1]. Chronic lesions cause serious pain and represent a considerable financial burden for many patients worldwide [2], and data from the World Health Organization indicate that every year, an estimated 300,000 deaths are caused by burns [3]. Patients with severe burns often suffer from metabolic, neurological and orthopedic complications, have psychosocial difficulties, and the time required for healing can be prolonged for months or years [4]. There is an urgent requirement for a good biological understanding of the mechanisms underlying wound repair, which can be simplified into four main phases: the hemostasis phase, the inflammation phase, the proliferation phase and the dermal remodeling stage. The pathophysiology of the burn wound is characterized by three injury zones, namely (i) the coagulation zone, which represents the tissue that was destroyed at the time of the injury, (ii) the stasis zone, which surrounds the coagulation zone and is characterized by inflammation and low levels of perfusion and (iii) the area of hyperemia. In most cases, the stasis area will progress and become necrotic 48 h after the thermal injury, and the initial burn will spread. The current management of patients with severe thermal injuries considers early wound closure, the prevention of septic complications and control of the external environment [5,6]. After burn injuries, there is an important loss of fluids and extensive tissue damage, and the thickness of the damaged dermal layer will have a significant influence on the wound-healing process [7]. Wound infection is a common complication, while systemic inflammatory and immunological responses might lead to a higher predisposition to sepsis, and the appropriate clinical treatments play a fundamental role in reducing mortality rates associated with this type of injury [7]. Unfortunately, there are many challenges regarding the antimicrobial activity of conventional topical treatments that are currently used to treat and prevent burn infections. Thus, several approaches have been used to improve the treatment of superinfected wounds with poor healing potential [8]. The preparation of new drug-carrying materials can be a promising alternative to these problems [1]. Because hydrogels have many advantages, such as non-toxicity and increased flexibility, they represent a wonderful option to improve wound healing and skin regeneration and can be a good absorbent of contaminated fluids, providing a favorable environment during the wound healing phases [9,10,11]. Structural similarity to the extracellular matrix, as well as good permeability, makes hydrogels inherently suitable for this type of application [12,13]. Hydrogels based on natural polymers present a low cost, have bioresorption ability, ensure good biocompatibility, show biodegradability and participate in supporting internal cellular activities [14,15]. Synthetic polymers have increased purity, the synthesis process is reproducible and they can release drugs in a controlled and sustained manner over a long period of time compared to natural polymers [16]. The incorporation of nanostructures (like nanofibers, nanoparticles, liposomes and micelles) into hydrogels aims to obtain advanced materials with improved properties (good cellular adhesion and good mechanical properties) [17,18]. In recent years, researchers have tried to develop new topical materials with antimicrobial effects to improve the wound healing process by minimizing pain and preventing infection, but, unfortunately, the results have not been encouraging. In this context, the present study was carried out with the aim of preparing a new antibacterial biocomposite hydrogel based on hyaluronic acid (HA) and sodium alginate (AG) containing ZnO nanoparticles and Ibuprofen with anti-inflammatory and antimicrobial action. The main novelty of the present study concerns the preparation of a biocomposite drug delivery system capable of effectively mimicking the extracellular matrix, which can exhibit improved antibacterial properties due to ZnO nanoparticles and can relieve pain without compromising wound healing while having anti-inflammatory properties due to the use of Ibuprofen.

HA is a linear non-sulphated polysaccharide, consisting of N-acetylglucosamine and glucuronic acid in the form of disaccharide repeats [19], that presents excellent biocompatibility. On the other hand, AG is a cationic polysaccharide obtained from brown algae that contain (1–4) linked β-D-mannuronic (M) and α-L-guluronic (G) acids. This biocompatible and non-immunogenic material is usually used in the form of sodium salt and has the ability, at ambient temperature, to form gels in contact with bivalent cations [20]. Bacterial resistance to antibiotics has become a major problem worldwide, and the use of non-antibiotic treatments represents a promising alternative. ZnO NPs have demonstrated good antimicrobial activity on a variety of Gram-positive and Gram-negative microorganisms [21,22,23]. The physicochemical properties of the obtained hydrogels were assessed by different techniques, such as SEM, FTIR and UV–Vis. In addition, the swelling degree in solutions with different pH values (pH 7.4 and pH 5.4), in vitro Ibuprofen release kinetics, cytotoxicity on normal human dermal fibroblasts and antimicrobial activity were also investigated.

## 2. Materials and Methods

### 2.1. Materials

High-viscosity alginic acid sodium salt (AG) was acquired from VWR International (Waltham, MA, USA), and Ibuprofen sodium salt ≥ 98% (Steinheim, Germany) and glycerine ≥ 99.7% (Schnelldorf, Germany) were acquired from Sigma-Aldrich. Zinc oxide nanoparticles with diameters between 10 and 30 nm (ZnO, 99+%,) were acquired from US Research Nanomaterials, Inc. (Houston, TX, USA). The materials required for the cytotoxicity assay, such as human dermal fibroblasts cell line (HDFa), 10% fetal bovine serum (FBS), non-essential amino acids, Dulbecco’s Modified Eagle Medium (DMEM), antibiotics (streptomycin/penicillin), phosphate-buffered saline (PBS), trypsin-EDTA and 95% sodium hyaluronate (HA) were purchased from Thermo Fisher Scientific (Waltham, MA, USA). The 4-(4,6-Dimethoxy-1,3,5-triazin-2-yl)-4-methylmorpholinium chloride (DMT-MM) necessary for the hydrogel synthesis was purchased from Merck Millipore (Darmstadt, Germany). Staphylococcus aureus-ATCC 25923 freeze-dried stains were obtained from ATCC (Manassas, VA, USA), and Chapman agar (mannitol salt agar) was acquired from Oxoid (Hampshire, United Kingdom).

### 2.2. Biocomposite Hydrogel Preparation

The hydrogels based on AG and HA were prepared by the condensation reaction of the two polymers, and a full-interpenetrated network was obtained. The esterification of the carboxylic groups with the hydroxyl groups took place in the presence of the activator DMT-MM (Figure 1). Three types of hydrogels, prepared in accordance with the experimental program presented in Table 1, were obtained as follows: (i) hydrogels obtained using the DMT-MM activator; (ii) hydrogels containing zinc oxide nanoparticles (ZnO nanoparticles) prepared by a condensation reaction using DMT-MM as the activator (biocomposite hydrogel); (iii) ibuprofen-loaded biocomposite hydrogel. The steps for the preparation of the hydrogels are presented below. The polymer solution was prepared by dissolving the appropriate amounts of AG and HA (according to the table) in 40 mL of distilled water at 50 °C using magnetic stirring. Separately, the DMT-MM activator was been solubilized in 5 mL of distilled water for 15 min at ambient temperature under gentle magnetic stirring. A suspension of ZnO nanoparticles was obtained by dispersing them in 5 mL of distilled water in an ultrasound bath. After good solubilization of the two polymers, a volume of 1.5 mL of glacial acetic acid was added, resulting in a solution with increased viscosity and a pH = 3.6. In the next step, the DMT-MM solution was introduced drop by drop, under strong magnetic agitation, to the polymer solution. At pH ˂ 4, the carboxylate groups in the alginate skeleton become protonated, and, first, hydrogen bonds are formed and, second, the esterification reaction of the formed carboxylic groups and the hydroxyl groups can take place using the activator. The biocomposite hydrogel was obtained through the homogenous distribution of the ZnO nanoparticle suspension into the hydrogel matrix, which was achieved by physically embedding the ZnO nanoparticles (Figure 2). For the drug-loaded hydrogels (AGAH-Z5-I-1; AGAH-Z5-I-2; AGAH-Z5-I-3), the amount of the activator was solubilized in 3 mL of purified water, and a volume of Ibuprofen solution (2 mL) at a concentration of 100 mg/mL was introduced into the polymer solution after the addition of the activator. The obtained mixture was then allowed to mix for 24 h at ambient temperature to favor the esterification reaction. Finally, to improve the elasticity of the system, a quantity of 500 mg of glycerine was introduced. The blend obtained was placed in a silicone mold with a side of 6.5 cm. The samples were dried in an oven at 50 °C. Afterwards, the hydrogels were washed with purified water and dried again until they reached a constant weight.

### 2.3. Characterization

#### 2.3.1. FTIR Spectroscopy

The characterization of the obtained materials from a structural point of view was carried out using FTIR spectroscopy. The FTIR analysis was carried out in absorbance mode with a wavelength in the domain 400–4000 cm^−1^ using a Shimaszu IRSpirit spectrometer equipped with a QATR™-S Single-Reflection ATR Accessory with a Diamond Crystal accessory.

#### 2.3.2. Hydrogel Microscopy

The obtained materials were analyzed superficially and in sections by SEM microscopy using a Vega Tescan microscope. This analysis revealed information about the degree of porosity and uniformity of the dry hydrogels.

#### 2.3.3. Swelling Behavior 

The swelling degree of the tested materials was studied to assess their behavior in skin wound environments. The gravimetric method was used to evaluate the swelling degree of the hydrogels in two solutions with different pH values, namely pH 7.4 and pH 5.4. For the experiment, square pieces of hydrogel were used with a weight of approximately 150 mg. The hydrogels were introduced into PBS solutions under gentle magnetic agitation at 37 °C for 24 h. The swollen hydrogels were weighed after one day. The water on the hydrogel was removed by dabbing it with filter paper. The equation below was used to determine the swelling capacity:(1)$$Q\left(\%\right)=\frac{W-W_{0}\;}{W_{0}}*100$$

*W* = mass of the swollen hydrogels;

*W*_0_ = mass of the hydrogels before being immersed in the environment.

#### 2.3.4. Ibuprofen Release Test

An in vitro release assessment was performed using static Franz diffusion cells (Copley vertical diffusion cells system) with an artificial cellulose membrane as a model membrane. The membrane separates the donor compartment containing the Ibuprofen-loaded hydrogels from the receiving compartment containing a volume of 7 mL release medium. For this test, two solutions with different pH values were used (a PBS solution with pH = 7.4 and a PBS solution with pH = 5.4; 37 ± 1 °C). Good agitation was ensured in the receiving compartment throughout the experiment. After defined times, 1 mL of medium with the drug was taken from the receiving compartment and replaced with the same volume of new prethermostated medium. The concentration of Ibuprofen in the receiving medium was determined using a Nanodrop One UV–Vis spectrophotometer at a wavelength of 223 nm. The release efficiency of Ibuprofen from the analyzed hydrogels (*R_ef_* (%) was calculated using the equation below:(2)$$Ref\left(\%\right)=\frac{mr}{ml}*100$$

*m_r_* = the quantity of Ibuprofen sodium salt released from the hydrogels (mg); *m_l_* = the quantity of Ibuprofen sodium salt loaded into the hydrogels (mg).

#### 2.3.5. Assessment of the In Vitro Cytotoxicity

The studied hydrogels, after sterilization for 3 min under UV radiation, were brought into direct contact with adult, adherent fibroblast cells of dermal origin—Human Dermal Fibroblasts adult cell line (HDFa). The cytotoxicity assessment of the hydrogels was carried out through the colorimetric method with MTT at 24 and 48 h after incubation. Each type of material was tested in triplicate, and the results were compared to the control sample. After the thawing step, the fibroblast cells were cultured in growth medium composed of DMEM, 10% FBS and an antibiotics cocktail using culture flasks (25 cm^2^ NuncTM EasYFlask TM with a total working volume of 7 mL). After the fibroblasts reached the desired confluence (70–80%), they were treated with 0.05% EDTA trypsin and incubated at 37 °C for 3–5 min. After checking the cell detachment under an inverted microscope (CKX41, OLYMPUS with image acquisition system (camera) and QuicKPHOTO Camera 3.0 software), the action of EDTA trypsin was blocked with growth medium, and then the cells were separated from the medium by centrifugation (1000 rpm for 5 min). After the separation and removal of the supernatant, the cell pellet was subcultured. The determination of the cell viability was performed on an automatic cell counter (EVE™ Automatic cell counter with EVE PC software) by trypan blue exclusion assay. One day after culturing the cells in 96-well plates, the growth medium was changed and the hydrogels were placed in contact with the fibroblast cells and incubated. At the end of the incubation time, the environment was changed to a fresh one (100 μL) and MTT solution (10 μL of 5 mg/mL), and, subsequently, the cells were kept at 37 °C for 4 h under humid conditions and 5% CO_2_ (incubator (air/CO_2_), MCO-5AC, Sanyo). After 4 h, another procedure consisted of removing 90 μL of the culture liquid and dissolving the formed formazan precipitate with 100 μL of DMSO solution and placing the cultures in the thermostat for 10 min. The samples were determined spectrophotometrically at λ = 570 nm on a Multiskan FC automated microplate reader and Skanlt Software 4.1.

#### 2.3.6. Antimicrobial Activity of Biocomposite Hydrogels

The antimicrobial activity of the hydrogels containing ZnO nanoparticles was evaluated against Staphylococcus aureus (Gram-positive bacteria) using the zone of inhibition test on agar plates. Culture media for Gram-positive bacteria (Chapman agar) was used to perform the antimicrobial tests. The sterilized and round-cut hydrogels (with a diameter of 4 mm) were humidified with purified water, introduced onto the plate and kept at 37 °C for 24 h. The areas around the material where microorganisms did not grow or survive were measured to establish the antimicrobial activity [24]. Each sample was sterilized using 70% ethanol in order to eliminate possible microbial contamination. At 24 h, the inhibition zone diameter was measured.

## 3. Results and Discussion

### 3.1. FTIR Spectroscopy

The obtained results presented in Figure 3 and Figure 4 and Table 2 confirmed the esterification reaction that took place between the AG and HA. The peaks present in the range of 3328 to 3391 cm^−1^, which appear in all the tested samples, can be assigned to O-H groups that reveal the presence of intermolecular and intramolecular hydrogen bonds. Also, the absorption band at about 2808 cm^−1^ characteristic of the –CH group can be distinguished. The absorption band at about 1738 cm^−1^ (the area surrounded by red in Figure 3 and Figure 4) can be assigned to the ester group (C=O), which appeared after the cross-linking of AG with HA. The incorporation of ZnO nanoparticles into the biocomposite materials was proved by the appearance of the absorption band at about 766 cm^−1^, which was assigned to the ZnO nanoparticles (the area surrounded by green in Figure 3 and Figure 4) [25,26].

### 3.2. Morphological Characterization

The morphology of the hydrogels with and without ZnO NPs is presented in Figure 5 and Figure 6. Cross-sectional microscopy of the AGAH-1 hydrogels demonstrates the presence of some fibrous bands that are generally characteristic of polysaccharides (AG). On the other hand, the AGAH-Z4-1 type hydrogels have a porous structure, the pores being small and non-uniform, and this change in the morphology of the hydrogels is mainly due to the addition of ZnO nanoparticles into the system. It can also be observed that the hydrogels without ZnO nanoparticles are smoother than those containing ZnO nanoparticles.

### 3.3. Hydrogel Swelling Capacity 

The activator quantity used in the synthesis (Figure 7) and the amount of ZnO nanoparticles (Figure 8) had a significant influence on the evolution of the swelling degree in aqueous environments. The volume of aqueous environment that penetrates the hydrogels is conditioned by the suppleness of the membrane, which may determine the diffusion rate.

As shown in Figure 7, the swelling degree of the hydrogels without HA but with different molar ratios between AG and the activator was between 948% and 434% for the hydrogels tested in an environment with pH = 7.4 and between 852 and 404% for the hydrogels tested in an environment with pH = 5.4. The results of the degree of swelling of the hydrogels with and without ZnO nanoparticles presented in Figure 8 indicate values between 403% and 107% for the hydrogels tested in the medium with pH = 7.4 and between 381% and 102% for the hydrogels tested in an environment with pH = 5.4. It was observed that at pH = 7.4, the swelling degree of the hydrogels was slightly higher. The carboxylate groups formed in an alkaline environment lead to electrostatic repulsions that will allow the infiltration of the aqueous medium inside the hydrogel and will lead to an increase in the degree of swelling. Moreover, as expected, it was found that with an increase in the activator amount, the swelling capacity of the hydrogels decreased, which was due to the augmentation in the cross-linking degree as a result of better reactivity between the two polymers in the presence of an increased quantity of the activator. Furthermore, the hydrogel swelling capacity was also strongly affected by the initial proportion between AG and HA. It was observed that as the quantity of HA from the polymer blend increased, the swelling capacity decreased. Increasing the amount of HA in the system will bring more carboxylic and hydroxyl groups, which leads to the formation of more ester groups and, implicitly, to an increase in the cross-linking density, which will ultimately lead to a decrease in the degree of swelling. Moreover, increasing the amount of HA brought more hydroxyl groups (OH) into the system, which were involved in intra- and intermolecular hydrogen bonds, which led to a decrease in the swelling degree of the hydrogels. Finally, increasing the amount of ZnO nanoparticles in the hydrogels resulted in a decrease in the swelling degree owing to a decrease in the available volume inside the hydrogel meshes.

The swelling degree data for the hydrogels obtained in the absence and presence of ZnO nanoparticles in solutions with pH = 7.4 and pH = 5.4 after 24 h were processed through the one-way ANOVA statistical test in order to evaluate the pH influence on swelling. The significance level between those two groups was 0.610, which was higher than the threshold value of 0.05, indicating that the pH did not have a significant influence on the swelling degree.

### 3.4. Ibuprofen Release Studies

Figure 9 shows a comparison between the in vitro release rate of Ibuprofen from the hydrogels without ZnO nanoparticles (AGAH-I-1) and those with ZnO NPs (AGAH-Z4-I-1, AGAH-Z5-I-1) over a period of 24 h. The release of the drug took place in two stages: in the first stage there is a slow release of the drug in the first 300 min, and in the second stage there is a rapid release until stability. The release efficiency varied between 68 and 95% and was higher in the case of the hydrogels without ZnO nanoparticles. It was found that an increase in the amount of ZnO nanoparticles led to a decrease in the drug release rate, which was expected due to a reduction in the meshes of the polymer network.

### 3.5. Cytotoxicity Assessment

It is very important to know the potential effects of the materials that are going to come into contact with the tissues of the human body. One of the most frequently used tests to evaluate the biocompatibility of biomaterials is the cytotoxicity test. Therefore, it is important that the use of a new material in the human body does not have side effects on the cells and surrounding tissues. Several types of hydrogels were selected to verify cytotoxicity, and the results are shown in Figure 10 and Figure 11. It was observed that the HDFa cells that were treated with materials containing ZnO nanoparticles (Figure 10) exhibited excellent viability, both at 24 and 48 h. These values varied between 91% and 98%, which clearly demonstrates that the tested hydrogels are non-cytotoxic.

On the other hand, following the treatment of the cells with hydrogels containing ZnO nanoparticles (Figure 11), a decrease in cell viability was observed. This behavior was also found in several studies in the literature [26]. The cellular viability after a day of treatment varied between 68.05% and 78.49%, demonstrating the fact that the tested hydrogels showed weak cytotoxicity. After 48 h of exposure, the cell viability varied between 45.15% and 60.27%, which proved the fact that the obtained biocomposite hydrogels show weak to moderate cytotoxicity.

When speaking about cell viability at 24 h versus 48 h, an ANOVA statistical analysis performed for the hydrogels with and without ZnO nanoparticles revealed values for a significance level of 0.31 and 0.15, respectively, indicating that it is more probable for hydrogels without ZnO nanoparticles to manifest a decrease in the cell viability over time.

### 3.6. Antimicrobial Activity of Biocomposite Hydrogels

Bacteria are generally made up of a cell membrane, a cell wall and cytoplasm, and the cell wall is outside the cell membrane and plays a role in maintaining the osmotic pressure of the cytoplasm as well as the characteristic shape of the cell. Gram-positive bacteria have a cytoplasmic membrane with a multi-layered peptidoglycan polymer [28] and a thicker cell wall of 20–80 nm. Nanoparticles with sizes in these ranges can easily pass through peptidoglycan and, therefore, can destroy the integrity of bacterial cells. Researchers are analyzing the bacterial morphological changes induced by ZnO nanoparticles, but the exact mechanism of toxicity still remains controversial, as there are some questions regarding the spectrum of antibacterial activity that require deep explanations. The mechanisms presented so far in the specialized literature are the following: the direct contact of ZnO nanoparticles with the cell wall, resulting in the destruction of the bacterial cell’s integrity [29] and the release of antimicrobial ions (Zn^2+^ ions) [30].

The results presented in Figure 12 and Table 3 show that the obtained biocomposite hydrogels have strong activity against Gram-positive bacteria (*S. aureus*). 

Four types of hydrogels, with different ratios between the polymers and different amounts of ZnO nanoparticles (namely AGAH-Z4-1 and AGAH-Z4-2, which contained 4% ZnO nanoparticles and AGAH-Z5-1 and AGAH-Z5-2, which contained 5% ZnO nanoparticles), were chosen. As expected, the samples with a higher content of ZnO nanoparticles demonstrated higher antimicrobial activity against the *S. aureus* strain. Also, it was found that the ratio between the polymers did not have a major effect on the antimicrobial activity. The obtained data are in concordance with those found in the specialized literature [26,31].

## 4. Conclusions

A series of biocomposite hydrogels, based on different ratios between hyaluronic acid (HA) and sodium alginate (AG), were prepared using DMT-MM as an activator. Ibuprofen (as a model anti-inflammatory drug) and ZnO NPs (at two concentrations) were both loaded. The esterification reaction between AG and HA in the presence of the DMT-MM activator was observed by FTIR. Moreover, the morphology of the hydrogels was investigated by SEM. The presence of ZnO NPs has an influence on both the morphology and the swelling degree, which varied between 100 and 950% as a function of the amount of the activator, HA and ZnO NPs. From drug release kinetics data, during 24 h, it was observed that the drug release ranged from 68 to 95%, as a function of the ZnO NPs and the cross-linking density. The cellular viability tests demonstrated the non-cytotoxic behavior of the hydrogels without ZnO NPs, whereas a weak to moderate cytotoxic effect was noticed for hydrogels with ZnO NPs. The antimicrobial activity was investigated against *S. aureus,* and it appeared that the hydrogels with a concentration of 5% of ZnO NPs have the highest antimicrobial activity. The obtained results are very encouraging and prove that further detailed in vitro and in vivo tests can be performed with the purpose of demonstrating the efficiency of these hydrogels as biomaterials suitable for the treatment of burn wounds.

## Figures and Tables

**Figure 1 pharmaceutics-15-02240-f001:**
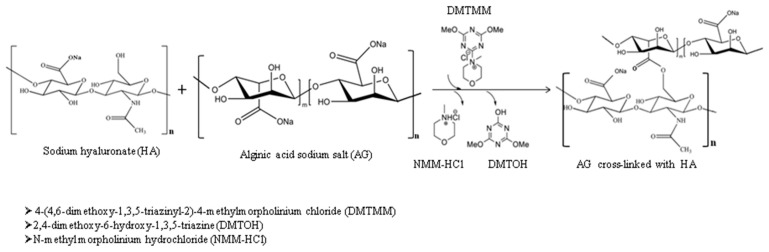
Reaction scheme of hyaluronic acid and sodium alginate in the presence of DMT-MM.

**Figure 2 pharmaceutics-15-02240-f002:**
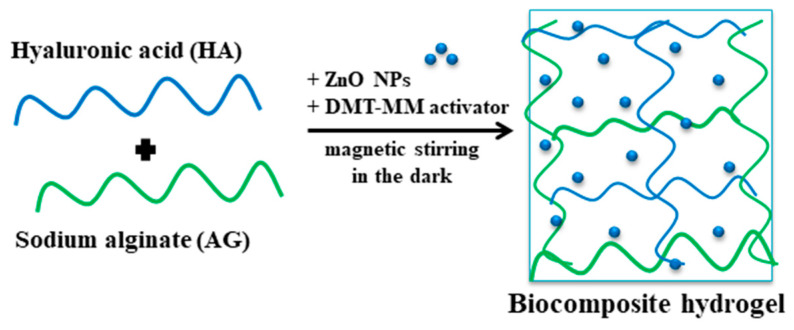
Schematic illustration of biocomposite hydrogels containing ZnO NPs.

**Figure 3 pharmaceutics-15-02240-f003:**
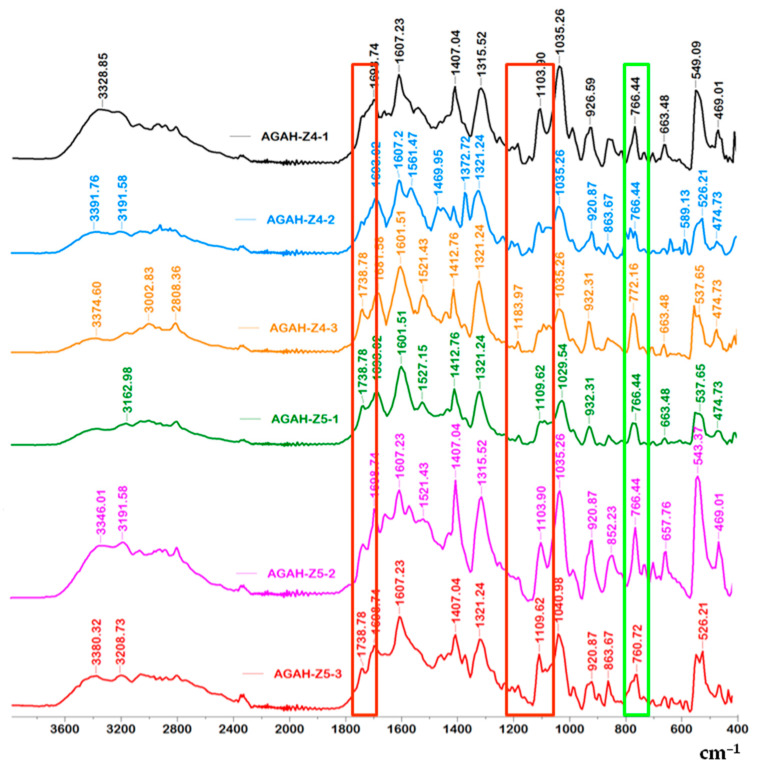
FTIR spectra for the biocomposite hydrogel samples containing ZnO nanoparticles.

**Figure 4 pharmaceutics-15-02240-f004:**
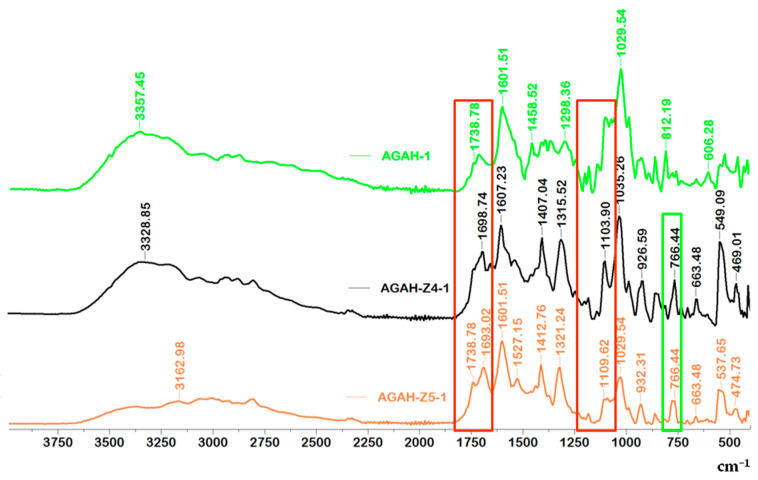
FTIR spectra for hydrogel samples with and without ZnO nanoparticles.

**Figure 5 pharmaceutics-15-02240-f005:**
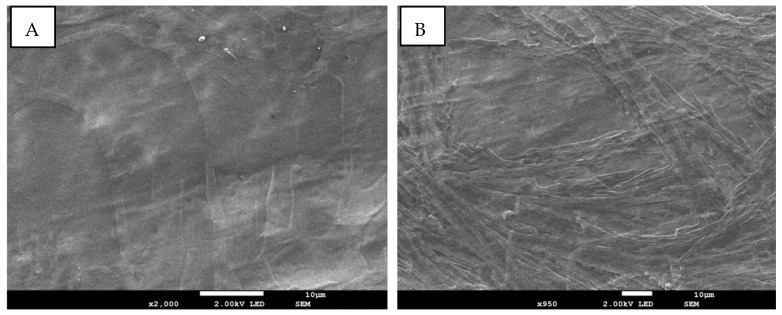
Surface (**A**) and cross-section (**B**) SEM micrographs of the hydrogel type AGAH -1 without ZnO nanoparticles.

**Figure 6 pharmaceutics-15-02240-f006:**
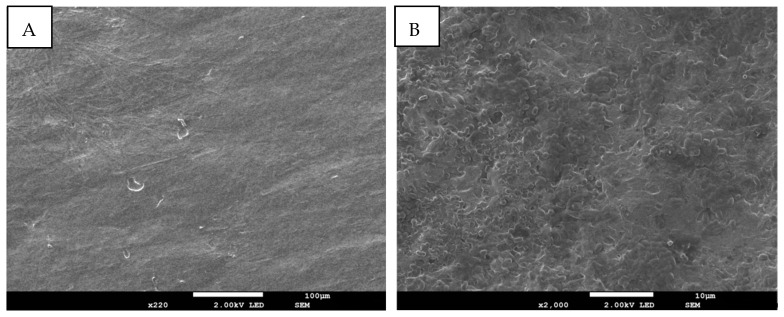
Surface (**A**) and cross-section (**B**) SEM micrographs of the hydrogel type AGAH-Z4-1 containing ZnO NPs.

**Figure 7 pharmaceutics-15-02240-f007:**
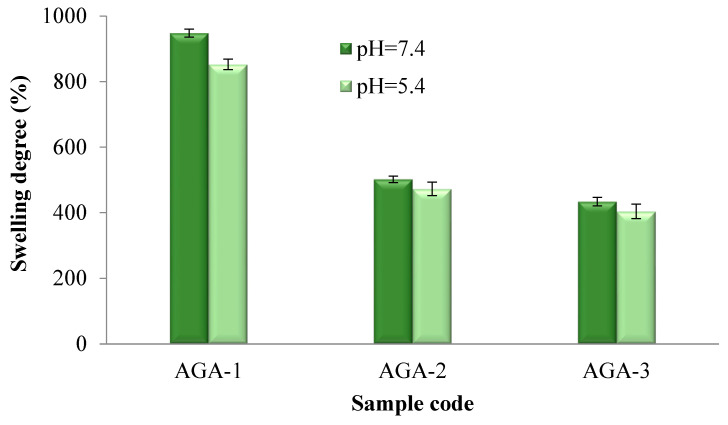
Swelling degree after 24 h in solutions with pH = 7.4 and pH = 5.4 for hydrogels obtained by varying the ratio between the activator and the polymers.

**Figure 8 pharmaceutics-15-02240-f008:**
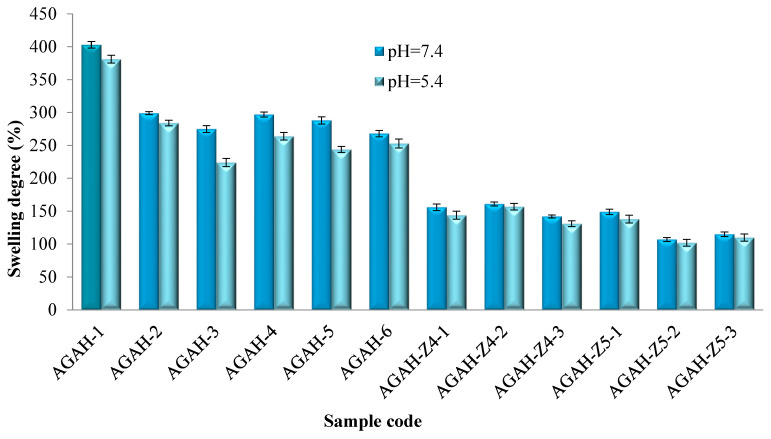
Swelling degree after 24 h in solutions with pH = 7.4 and pH = 5.4 for hydrogels obtained in the absence and presence of ZnO nanoparticles.

**Figure 9 pharmaceutics-15-02240-f009:**
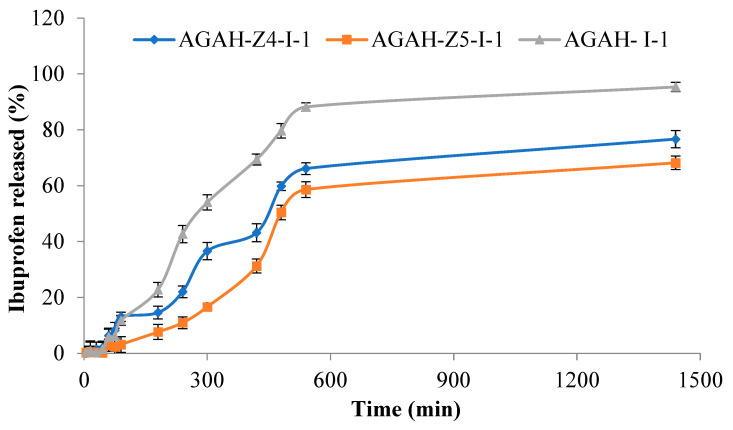
In vitro release profiles of Ibuprofen from hydrogels with and without ZnO nanoparticles.

**Figure 10 pharmaceutics-15-02240-f010:**
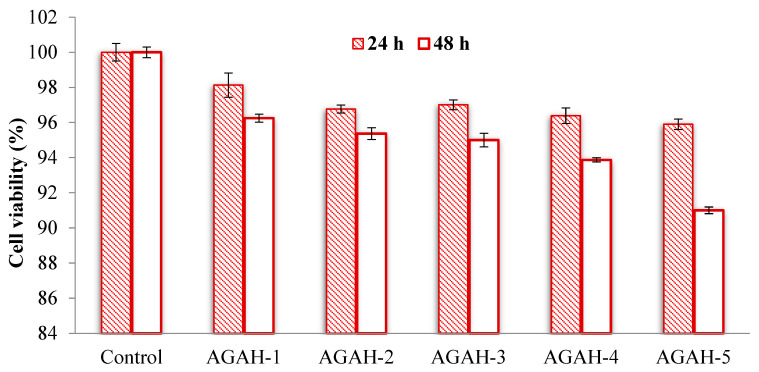
Evolution of cell viability after 24 and 48 h of treatment with hydrogels without ZnO nanoparticles.

**Figure 11 pharmaceutics-15-02240-f011:**
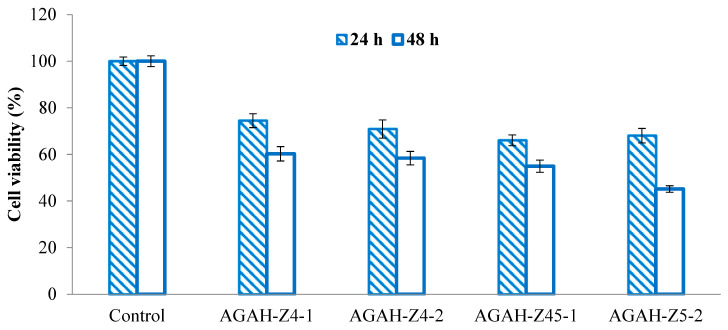
Evolution of cell viability after 24 and 48 h of treatment with biocomposite hydrogels containing ZnO nanoparticles.

**Figure 12 pharmaceutics-15-02240-f012:**
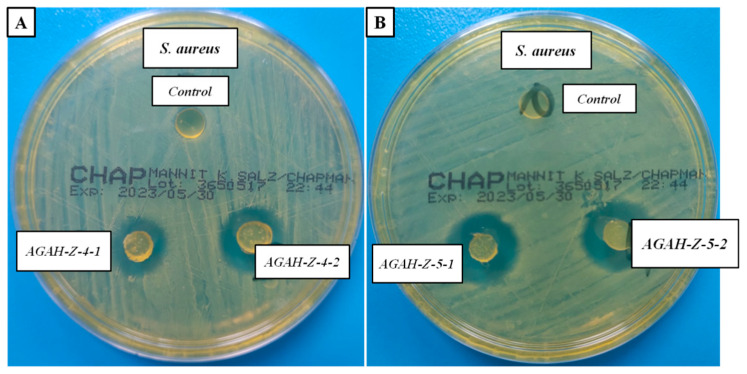
Antimicrobial activity of biocomposite hydrogels containing 4% ZnO nanoparticles (**A**) and 5% ZnO nanoparticles (**B**) against *S. aureu*.

**Table 1 pharmaceutics-15-02240-t001:** Experimental program used to obtain hydrogels.

Sample Code	AG/HA Ratio (mg/mg)	Moles DMT-MM/Moles COOH	ZnO NPs Compared to the Polymer Amount (%)	Ibuprofen (mg)
AGA-1	100/0	2/1	-	-
AGA-2	100/0	3/1	-	-
AGA-3	100/0	3.5/1	-	-
AGAH-1	90/10	3/1	-	-
AGAH-2	80/20	3/1	-	-
AGAH-3	70/30	3/1	-	-
AGAH-4	90/10	3.5/1	-	-
AGAH-5	80/20	3.5/1	-	-
AGAH-6	70/30	3.5/1	-	-
AGAH-Z4-1	90/10	3/1	4	-
AGAH-Z4-2	80/20	3/1	4	-
AGAH-Z4-3	70/30	3/1	4	-
AGAH-Z5-1	90/10	3/1	5	-
AGAH-Z5-2	80/20	3/1	5	-
AGAH-Z5-3	70/30	3/1	5	-
AGAH- I-1	90/10	3/1	-	200
AGAH-Z4-I-1	90/10	3/1	4	200
AGAH-Z5-I-1	90/10	3/1	5	200

**Table 2 pharmaceutics-15-02240-t002:** Characteristic adsorption bands of hydrogels with and without ZnO nanoparticles.

Wavenumber (cm^−1^)	Functional Group	References
766	vibration band of ZnO	[25,26]
812–932	guluronic and mannuronic acids from AG	
1035	C-O-C functional group from AG	[27]
1103–1183, 1738	ester bonds (C=O and C–C(O)–C)	
1521–1561	amide I and amide II bonds from HA	
1601, 1607	asymmetric vibrations of the COO^–^	
1407, 1412	symmetric COO^−^ vibration	
1693, 1698	C=O asymmetric stretching of the amide group	
2808	–CH vibration	
3328–3391	O-H groups	

**Table 3 pharmaceutics-15-02240-t003:** Inhibition indices of biocomposite hydrogels against *S. aureus*.

Inhibition Zone (mm)	
Samples	Gram-Positive Bacteria *S. aureus* (ATCC 25923)
AGAH-1 (Control)	no inhibition zone
AGAH-Z4-1	18
AGAH-Z4-2	20
AGAH-Z5-1	22
AGAH-Z5-2	23

## Data Availability

The data presented in this study are available on request from the corresponding author.

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
