# Peer review of "Influence of ZnO Nanoparticles on the Properties of Ibuprofen-Loaded Alginate-Based Biocomposite Hydrogels with Potential Antimicrobial and Anti-Inflammatory Effects"

_pharmaceutics, 2023, doi:10.3390/pharmaceutics15092240_

Round 1

Reviewer 1 Report

There are the following comments and recommendations on the work:

Introduction

Add references to new studies dated 2022-2023. In particular, one should consider works on various polymer gels for drug delivery, wound healing, nanoparticles of metals and metal oxides as antimicrobial agents (doi: 10.3390/pharmaceutics15030830, 10.3390/polym15132831, 10.3390/gels8060329, 10.3390/ polym14050864).

Why did the authors of the work use zinc oxide nanoparticles as an antimicrobial agent, and not other nanoparticles, for example, based on silver? Provide a detailed explanation. There are works devoted to hydrogels based on sodium alginate and zinc oxide nanoparticles that have antibacterial activity against Escherichia coli, Staphylococcus aureus, Candida albicans and methicillin-resistant S. aureus (MRSA) and non-toxic against human dermal fibroblasts (see. for example, the work http://dx.doi.org/10.2147/IJN.S79981 ). What is the scientific novelty of the work? Explain why the authors used hyaluronic acid and not other polymers such as chitosan, which is also biocompatible.

Add to the text of the Introduction that hyaluronic acid consists of N-acetylglucosamine and glucuronic acid in the form of disaccharide repeats.

Materials and Methods

Specify the country of origin of the reagents. Give the reaction scheme of hyaluronic acid and sodium alginate in the presence of an activator. Indicate the pore size of the membrane, the diameter and volume of the static diffusion Franz cells.

Results and discussion

On the IR spectra, it is necessary to present the wavenumber values only for those signals that are described in the text of the article and confirm the occurrence of the esterification reaction and the presence of zinc oxide nanoparticles in the gel.

How does the initial ratio of AG and HA affect the degree of swelling of the gel and why? Give the necessary explanations for Figure 7.

Add a link to Figure 10 (see sentence This behavior was also found in several literature studies).

Figure 11 (right figure) shows no results for the control sample. Give the mechanism of the antimicrobial action of the resulting gel containing zinc oxide nanoparticles. Correct the name of the bacterium to S. aureus (Table 2). Did the resulting gels exhibit antibacterial activity against P. aeruginosa?

Conclusions

The presence of zinc oxide nanoparticles adversely affects the viability of skin fibroblasts (HDFa cell line). What is the reason for this? Provide a detailed explanation. How do the authors plan to increase cell viability to the required level (over 70%)?

English needs minor editing

Reviewer 2 Report

Overview of the manuscript
The work is focuses on the creation of a hydrogel capable of performing anti-inflammatory and antibacterial properties, thanks to substances as ZnO and Ibuprofene. The structural properties of the hydrogels and its ability to act as a therapeutic adjuvant has been analysed.

 GENERAL COMMENT

The work is interesting, and it well addressed on the mainstream studies dealing with analysis of several hydrogels to be used in therapeutic protocols, for the healing of different cutaneous lesions. The experimental plan is well structured, and the methodology adopted are appropriate and gives solidity to results and conclusions. However, the work should be ameliorated in its presentation, because several points are not well explained and remain confusing. The absence of any statistical test represents the main limitation of the study.

  SPECIFIC COMMENTS

Abstract

Pag. 1, line 9: “Ibuprofen, as a model drug”. The indication is confounding, in your experimental plan, Ibuprofen is a reference drug, but also a drug whose anti-inflammatory action is analysed together the antimicrobial action of the gels. Change the sentence and explain better.

Material and Methods

Pag. 3, line 35: It is not clear. Which type of experimental groups do you indicate here? Do they represent the repetition of the same experimental group, or are they three different experimental groups? These experimental groups cannot be found in Table 1.

Pag. 4, line 14-15: have you measured the hydrogel swelling in two different solution presenting two different point of pH? Be clearer.

Pag. 4, line 18: “predetermined time”? At what time? Be more specific.

Pag. 5, line 4-5: the sentence is not clear. The medium is a collection of two different solution with different pH, how is this possible? Explain better.

 Have you performed any statistical test?

 Results and Discussion

Fig. 6 and Fig. 7: the figures show bars without any error standard o deviation standard bars. Why? Aren’t they the results of repeated measurements?

Fig. 7: the impression is that in some experimental groups there is differences in swelling degree between pH 7.4 and pH 5.4. A statistical evaluation is important to identify differences between the different gel formulations.

Pag. 8, line 7: the percentage values do not seem the same presented in the graph. Which data are they? Explain better.

Pag.10, line 13: reference is lacking.

Fig. 9 and Fig. 10: a statistical test application should be appropriate, to consolidate the differences.

Fig. 11. Insert marks A and B. In the left picture, the target control hides the control gel.

In the two pictures the inhibition halo around the control gels does not appear to show a diameter of 9 mm, as stated in Table 2. Explain

Reviewer 3 Report

The manuscript is in principle very interesting. The authors study the use of new anti- bacterial biocomposite hydrogels based on hyaluronic acid (HA) and sodium alginate (AG), at different ratios, in the presence of 4-(4,6-dimethoxy-1,3,5-triazinyl-2)-4- methylmorpholinium chloride activator. where the anti-inflammatory and antimicrobial properties were induced by the loading of Ibuprofen, as a model drug, and different amount of zinc oxide nanoparticles (ZnO NPs) in wounded. The authors conclude that this treatment can be on interest as biomaterials for the treatment of burn wounds.

I have only one question:

Since the argument of burn injuries is not familiar for many people, may the authors better introduce the topic describing also the cellular mchanism of this kind of wound?

Moderate editing of English language required

Reviewer 4 Report

- The abstract must rewrite because there is no significant result of this research written in it.

- What is the novelty of this research?

- how about MIC and MBC tests in antimicrobial tests for this research?

- In Figure 2, the graph is showing transmittance or absorbance? The discussion for FTIR is incomplete. As there are many peaks and many samples, it is suggested to make a Table and show the peaks and functional bonds separately.

- Explain for Figures 2, and 3 what is the difference between FTIR graphs for "with ZnO" and "without ZnO" samples?

- The following references can be helpful for improving the discussion of the manuscript: https://doi.org/10.3390/polym15020272 , https://doi.org/10.1016/j.polymertesting.2020.106922.

- Figure 4, and 5 is incomplete, it is suggested to add the SEM image of ZnO nanoparticles. Also, explain well the difference between Figures 4, and 5.

- Figure 6 needs an error bar. The swelling test needs more than two times replications.

Round 2

Reviewer 1 Report

The authors corrected the manuscript in accordance with the recommendations and provided reasoned answers.

I recommend the manuscript for publication.

English needs minor editing.

Author Response

Thank you for the appreciation.

Reviewer 2 Report

The concerns previously raised have been addressed, however the statistical analysis has not been included in the bar-graph of figures 8, 10 and 11, although the authors express quantitative concepts. This issue should be fixed.

Reviewer 4 Report

The manuscript has been upgraded and all the comments have been addressed well. The manuscript can be accepted to be published in its current state.

Author Response

Thank the reviewer for the appreciation.